# Fast Stray Light Performance Evaluation Based on BSDF and Radiative Transfer Theory

**DOI:** 10.3390/s23229182

**Published:** 2023-11-14

**Authors:** Chaoli Zeng, Guangqing Xia, Xing Zhong, Lei Li, Zheng Qu, Qinhai Yang, Yuanhang Wang

**Affiliations:** 1State Key Laboratory of Structural Analysis, Optimization and CAE Software for Industrial Equipment, Dalian University of Technology, Dalian 116024, China; 2150126542@mail.dlut.edu.cn (C.Z.); gq.xia@dlut.edu.cn (G.X.); 2Key Laboratory of Advanced Technology for Aerospace Vehicles of Liaoning Province, Dalian University of Technology, Dalian 116024, China; 3Collaborative Innovation Center of Micro & Nano Satellites of Hebei Province, North China Institute of Aerospace Engineering, Langfang 065000, China; 4Chang Guang Satellite Technology Co., Ltd., Changchun 130102, China; 5Changchun Institute of Optics, Fine Mechanics and Physics, Chinese Academy of Sciences, Changchun 130033, China

**Keywords:** BSDF, radiative transfer theory, PST, off-axis reflective system, rapid quantitative evaluation

## Abstract

Evaluating the stray light cancellation performance of an optical system is an essential step in the search for superior optical systems. However, the existing evaluation methods, such as the Monte Carlo method and the ray tracing method, suffer from the problems of vast arithmetic and cumbersome processes. In this paper, a method for a rapid stray light performance evaluation model and quantitatively determining high-magnitude stray light outside the field of view are proposed by adopting the radiative transfer theory based on the scattering property of the bidirectional scattering distribution function (BSDF). Under the global coordinates, based on the derivation of the light vector variation relationship in the near-linear system, the specific structural properties of the off-axis reflective optical system, and the specular scattering properties, a fast quantitative evaluation model of the optical system’s stray light elimination capability is constructed. A loop nesting procedure was designed based on this model, and its validity was verified by an off-axis reflective optical system. It successfully fitted the point source transmittance (PST) curve in the range of specular radiation reception angles and quantitatively predicted the prominence due to incident stray light outside the field of view. This method does not require multiple software to work in concert and requires only 10^–5^ orders of magnitude of computing time, which is suitable for the rapid stray light assessment and structural screening of off-axis reflective optical systems with a good symmetry. The method is promising for improving imaging radiation accuracy and developing lightweight space cameras with low stray light effects.

## 1. Introduction

In the pursuit of high resolution, compact size, and light weight, the radiation quality requirements of space optical payloads have increased [1,2,3,4]. Stray light is one of the major aspects impacting the performance of the optical payload [5,6,7]. Stray light elimination capability is becoming increasingly important [8]. The traditional measure for evaluating the stray light suppression capability of optical systems is to analyze the effect of stray light when imaging light at different off-axis angles after modeling the optical system with mechanical structures. The calculation methods of stray light include the Monte Carlo method, ray tracing method, zonal method, optical density method, and paraxial approximation method [9]. The Monte Carlo and ray tracing methods are relatively mature and universal. TracePro is an optical analysis program that combines the Monte Carlo method and ray tracing method [10]. The process requires a million ray scale analysis, which is time-consuming, tedious, and computationally intensive.

The design of internal and external stray light suppression structures, such as baffles, is based on the structural characteristics of the optical system, and the placement of the structure increases the volume and mass of the space optical payload [11]. Therefore, an optical system with a good stray light elimination capability is the focus of achieving lightweight miniaturization and improving imaging quality. It is important to study systems with a high stray light elimination capability in their optical system characteristics to find an optical system structure with a relatively superior stray light elimination performance in the same optical system imaging metrics [12,13]. Also, an optical system structure with superior stray light elimination performance can reduce the requirement for a quality mirror surface, reducing the expenditure. Furthermore, the rapid quantitative evaluation of the optical system’s stray light elimination performance is an important step and also has important scientific and engineering significance.

PST is frequently used to evaluate the ability of an optical system to suppress stray light outside the field of view. Traditionally, PST curves are plotted using the TracePro7.0 software in conjunction with program software, which is more time-consuming and complex. Based on the radiative transfer theory and Harvey–Shack scattering theory, this paper constructs a PST model by the specific structural properties of the off-axis reflective optical system and the scattering property of the specular surface, conducts the rapid quantitative evaluation of the ability of optical systems to suppress internal and external stray light, and effectively replaces the trace evaluation method of millions of rays within the range of the specular radiation reception angle. The determination condition for direct-incident stray light and once-scattered attenuation stray light is proposed, and the PST evaluation with the prediction of the peaks of strong stray light interference protrusions, such as the direct-incident stray light and the once-scattered attenuation stray light, can be realized by the loop-nested program.

## 2. Basic Theory and PST Research History

### 2.1. Mirror BRDF Properties

BSDF is a measure that describes the intensity of light scattered in different directions on the same surface, characterizing the spatial distribution of light scattered from a given surface. As illustrated in Figure 1 below. It is defined as the ratio of the radiance in the output direction to the irradiance in the input direction [14].
(1)$$\mathrm{BSDF}=\rho (\theta_{0},\phi_{0};\theta_{S},\phi_{S})=\frac{L(\theta_{S},\phi_{S})}{E(\theta_{0},\phi_{0})}$$
where *θ*_0_ and *ϕ*_0_ are the elevation and azimuth angles of the incident ray, respectively; *θ_s_* and *ϕ_s_* are the elevation and azimuth angles of the scattered ray, respectively; *L* is the differential radiance of the scattering surface; and *E* is the differential incident irradiance [15]. When the BSDF of the mirror of an optical system is expressed as a directional cosine function, its magnitude is only related to the direction of reflection and the direction of scattering [16].

The ABg model [17] is frequently used to fit BSDF data, and it is expressed as follows:(2)$$\mathrm{BRDF}=\frac{A}{B+{\left|\sin\theta_{s}-\sin\theta_{0}\right|}^{g}}$$
where *A*, *B*, and *g* are the parameters to be determined; *θ*_s_ is the angle between the scattering direction and the surface normal direction; and *θ*_0_ is the angle between the reflective direction and the surface normal direction.

### 2.2. Radiative Transfer Theory

Radiative transfer theory is the theoretical basis for eliminating stray light. It can be described as the process by which light reaches the surface of one object and transfers some of its energy to the surface of another object [18]. As illustrated in Figure 2 below.

Assuming that the power incident on the surface *A_S_* is *P_S_*, the power scattered to the receiving surface *A_C_* is
(3)$$P_{C}=\pi \cdot P_{S}\cdot \mathrm{BSDF}\cdot GCF$$
where *GCF* is the geometric configuration factor [19,20].
(4)$$GCF=\frac{A_{c}\cos\theta_{s}\cos\theta_{c}}{\pi R_{sc}^{2}}$$

The projected solid angle [21] divided by *π* is often called the *GCF*, so that
(5)$$\Omega =GCF\cdot \pi =\frac{A_{c}\cos\theta_{s}\cos\theta_{c}}{R_{sc}^{2}}$$
(6)$$P_{c}=P_{s}\cdot \mathrm{BSDF}\cdot \Omega$$

According to the radiative transfer theory, each of the sensor’s internal collector elements then becomes a new source of radiation. The radiation transfer calculation is performed once again from these new sources to subsequent collectors, and so on, until the stray photons reach a detector [12]. Thus, the sum of the stray radiation contributions from each mirror can be used to express the optical system’s overall stray radiation.

### 2.3. PST

PST equals the average detector irradiance *E_det_* divided by the incident irradiance *E_inc_* [22].
(7)$$\mathrm{PST}(\theta )=\frac{E_{det}(\theta )}{E_{inc}(\theta )}=\frac{P_{det}(\theta )/A_{det}}{P_{inc}(\theta )/A_{inc}}$$

In 1989, the Massachusetts Institute of Technology investigated the use of a PST model based on the BSDF combined with the structural parameters of the optical system to predict the stray light elimination performance of the optical system, instead of using ray tracing. These are the models [23]:(8)$$\mathrm{PST}=\left[f_{1}+f_{2}+f_{3}+f_{4}\right]\frac{\pi }{4F^{2}}$$

And
(9)$$\mathrm{PST}=\left[f_{1}(\theta )S_{1}(\theta )+f_{2}(\theta )S_{2}(\theta )+f_{3}(\theta )S_{3}(\theta )+f_{4}(\theta )S_{4}(\theta )\right]\frac{\pi }{4F^{2}}$$
where *F* is the F-number of the optical system; the *f* functions are the mirror BRDFs; and the *S* functions are the shading functions. The predicted PST closely matches the earlier predicted PST at small off-field angles and shows the expected asymmetry at larger angles. The PST model cannot predict peak mutations. Similarly, the PST model for coaxial optical transmission systems is as shown in Equation (4) [21,24]:(10)$$\mathrm{PST}=1-\frac{1}{4}\left|5(1-\left|\frac{\theta }{\theta_{max}}\right|)-1+\left(\frac{\theta }{\theta_{max}}\right)\right|$$
where *θ_max_* is determined by the lens length and aperture. In 2004, Gary L. Peterson quantitatively predicted the stray light energy based on the BSDF but limited it to stray light in the field of view due to using optical system invariants that are not a function of the angle of incidence of light [25]

Off-axis reflective optical systems have the advantages of long focal lengths and large fields of view and are widely used in space optical payloads [26,27]. When the above model is no longer applicable, the off-axis quantities change the angle of incidence of the incident light concerning the mirror, so that the light cannot be positively incident on the mirror. According to the Harvey–Shack scattering theory, the differences brought about by the different off-axis quantities are reflected in the different absolute magnitudes of the scattered intensity. Based on radiative transfer theory and BSDF, we developed a new PST model to achieve the optical system’s fast stray light performance evaluation.

## 3. Modeling of PST Based on Radiative Transfer Theory

Different angles of incident light produce different angles of outgoing light on the mirror because the optical system was determined previously, i.e., the position of the radiation-receiving surface relative to the emitting surface is determined. According to the Harvey–Shack scattering theory, different values of the scattering angle corresponding to the receiving surface result in different values of the received radiation. Therefore, the model of stray radiation transmission in off-axis reflective optical systems can be established based on light vector transmission.

### 3.1. Transfer Modeling of Off-Axis Reflective Optical Systems with Decenter and Tilt Characteristics

The variation in the light vector of a coaxial near-linear system in the near-axial region can be described by the transfer matrix theory, and off-axis systems are categorized as aperture off-axis or field-of-view off-axis systems, in which the decenter and tilt of the mirror surface affects the transfer of the light vector, and the specific relationship is deduced as Figure 3:
(11)$$\varphi \prime =2\beta =2\frac{-y}{r}$$
where *y* is the value of the decenter and *r* is the radius of the curvature. The effect of the decenter on the light vector is a change in the outgoing angle. Similarly, the effect of the tilt on the light vector is a change in the outgoing angle. As illustrated in Figure 4 below.
(12)φ′=2α
where *α* is the value of the tilt. Combined with the linear variation relationship of the light vector, the change in the light vector after passing through the *i*th mirror with decenter and tilt can be expressed as

(13)$$\left\{\begin{array}{ll} h_{i}=h_{i-1}+d_{i-1}\tan(\varphi_{i-1}) & i\neq 0\\ h_{0}=h_{incdence} & i=0\\ d_{0}=L & i=0\\ \varphi_{i}=-\frac{2}{r_{i}}h_{i}-\varphi_{i-1}+(-\frac{2}{r_{i}})y_{i}-2\alpha_{i} & i\neq 0\\ \varphi_{0}=\varphi_{incdence} & i=0 \end{array}\right.$$
where *h_incden__ce_* is the height from the chief ray of the light source-setting surface to the main optical axis of the system, *L* is the distance from the light source to the primary mirror, *φ_incden__ce_* is the angle of the incident chief ray, and *d* is the distance between the mirrors.

### 3.2. Energy Transfer Limitations: Radiation Acceptance Coefficient

When the source plane is used as the emitting plane and the primary mirror is used as the radiation-receiving plane, it does not correspond to stray light radiative transfer. Therefore, this process is described by the radiation-receiving coefficient (*S*) of its energy transfer. The light of the source surface is incident on the primary mirror at different angles, and the effective receiving area of the primary mirror receives a different energy, resulting in a different stray light energy. Similarly, the longer the length of the baffle, the stronger the limiting effect on the incident stray light at angles outside of the field of view. Therefore, the received energy can be expressed as a function of the angle of incident light. As illustrated in Figure 5 below. It can be simplified as shown in the following model, where *D* is the size of the light source surface/the size of the baffles’ aperture, *L* is the distance from the light source surface to the primary mirror/the length of baffles, and the axis is the primary mirror’s optical axis. When *D* = *d*, the maximum specular radiation reception angle *θ*_max_ can be expressed as
(14)$$\theta_{\max}=\tan^{-1}(\frac{d}{L})=\tan^{-1}(\frac{D}{L})$$

The radiation reception coefficient can be approximated as
(15)$$S(\varphi )=1-\frac{\sin(\varphi )}{\sin(\theta_{\max})}$$

### 3.3. Stray Light Passing through a Line Segment

The stray light suppression structure cannot be set up in the normal light imaging path, and some of the light outside of the field of view will either reach the mirror through the normal optical imaging path and then affect the detector, or be directly incident on the detector focal plane. This direct incident stray light or once-scattered attenuation stray light energy level is large; thus, its impact on the detector cannot be ignored.

This thesis proposes a determination condition to assess whether a certain incident angular degree of light causes a strong stray light effect under a defined optical system and the height of the chief ray at the source plane.

As it can be seen in Figure 6, the area enclosed by the triangle is the path of the normal imaging light and cannot be obscured. The triangle is enclosed by the intersection (point j) of the lower ray emanating from the *j*th surface with the upper ray emanating from the *j* + 1th surface. This triangular range can also be extended according to the structure of the optical machine. All points can be calculated according to the optical system design parameters or derived directly from the optical system model. Line *j* is the line connecting point *j* to point *j* + 1, the auxiliary line *j* is the line connecting the upper point of surface *j* to the lower point of line *j*, and the auxiliary line *j*’ is the line connecting the lower point of line *j* to the lower point of the surface where the light reaches. As shown above, when the slope of the angle of incidence is greater than the slope of line1, lower than the slope of the auxiliary line1, and lower than the slope of auxiliary line1′, i.e., within the angle *θ*, it represents the range in which the stray light generated by light from outside of the field of view arriving at the tertiary mirror causes the detector to respond. Also, the angle *θ* limits the through energy, using *θ*/90 to approximate the ratio of the through energy. The dashed black line represents the light that reaches the tertiary mirror directly, which produces a once-scattered attenuation stray light that causes the detector to respond, and this angle produces a PST peak.

Therefore, the determination condition can be described as follows: the condition for an emitted ray from the *j*th surface to reach the *j* + 3th-receiving surface is that the slope of the ray is greater than the *j*th line and lower than the *j*th auxiliary line (note that the light source surface is included, and the slope is calculated in absolute values).

As shown in the figure above, the stray light reaching the detector surface directly from the primary mirror must pass through the angular range formed by line2, auxiliary line2, and auxiliary line2′. According to this determination condition, the light passing through the primary mirror of this system cannot reach the detector directly. This optical system structure form has the suppression of the stray light directly incident on the detector from the primary mirror.

### 3.4. Modeling of PST Based on Radiative Transfer Theory

According to the above model, combining the BSDF of each mirror and the Ω of the optical system, the PST function model for the study of stray light radiation transmission can be established in global coordinates based on the light vector transmission model of the off-axis system. The BSDF of each mirror and Ω are a function of the angle. The calculation of the BSDF and Ω requires the derivation of the angles of reflection and scattering produced by light at different angles of incidence. The specific derivation process is as Figure 7:

The relationship between the angle of incidence *φ_i_*_−1_ on the *i*th surface and the angle of reflection *θ_i_*_0_ produced on the *i*th surface is as follows:(16)$$-\varphi_{i-1}+\theta_{i0}+\theta_{i0}=\varphi_{i}$$

Since *φ_i_* is the angle of the light after specular reflection, it is inconvenient to calculate it in a sequential iteration. According to the geometric relationship, it can also be expressed as
(17)$$(d_{i-1}-r_{i}+h_{i-1}÷\tan\varphi_{{}_{i-1}})\sin\varphi_{{}_{i-1}}=r_{i}\times \sin\theta_{i0}$$

The decenter can be considered as a shift in object height, and the tilt is equivalent to an angular change in the normal vector of the mirror. Therefore, the formula can be transformed as
(18)$$(d_{i-1}-r_{i}+(h_{i-1}-y_{i})÷\tan(\varphi_{i-1}+a_{i}))\sin(\varphi_{i-1}+a_{i})=r_{i}\times \sin\theta_{i0}$$

In a normal imaging path, the imaging light is reflected sequentially according to the designed optical system, and the receiving surface receives the normal light transferred from the previous surface. At this time, the reflection angle in the BSDF function and the scattering angle have the same value, and the value of the BSDF function reaches the maximum, characterizing the stray radiation of the optical system (determined by the roughness of the mirror surface and other factors). The hemispherical integral of the BSDF (total integrated scatter) also represents the ratio of the total energy of the scattered light to the total energy of the incident light, which is proportional to the square of the root-mean-square roughness. Therefore, with a rougher specular surface, the generated stray light energy increases exponentially.

If the incident light is outside of the field of view, the angle of reflection in the BSDF function is not equal to the angle of scattering, according to the theory of vectorial ray transmission. As shown below, the green dotted line is the normal incident light (*h*_i+1′_), and the blue line is the non-normal imaging ray. In this paper, to achieve the rapid evaluation of PST and a reduction in the program arithmetic, the transmission of a single ray (the chief ray of the source plane) was taken to represent the incident ray of the whole light source plane. Therefore, the angle *θ_is_* between the green dotted line and the normal direction of the mirror was defined as the scattering angle of the stray radiation transferred to the next mirror.

The *θ_is_* produced by the incident angle *φ_i_*_−1_ after reflection on the *i*th surface is as Figure 8:
(19)$$\theta_{\mathrm{is}}=\left|\left|\varphi_{i}\right|-\theta_{i0}-\arctan(\frac{\left|d_{i}\right|}{\Delta h_{i+1}})\right|$$

The value of each parameter of the Ω of the radiant energy transmitted through the *i*th surface and received by the *i* + 1th surface can be obtained from geometrical relations.
(20)$$\Delta h_{i+1}=h_{i}-h_{i+1}^{\prime }$$
(21)$$\theta_{ci}=\left|\varphi_{i}+\alpha_{i+1}\right|$$
(22)$$R_{sci}^{2}=d_{i}^{2}+\Delta h_{i+1}^{2}$$

The specular position (*h*_i+1′_) reached by the incident light within the normal field of view can be calculated from the light vector transmission of the normal light and stored for the calculation of PST values at different off-axis angles.

The expressions for Ω and BSDF based on the structural parameters of the optical system in terms of the angle of incidence as a variable were obtained by integrating Equations (13) and (19)–(22) into Equation (5), and Equations (13), (16) or (18), (19) and (20) into Equation (2).

Based on the above model, the PST can be expressed as a function of the incident angle as a variable based on the structural parameters of the optical system (radius, mirror distance, and mirror scattering properties). The total stray radiation is derived by summing the radiation from each mirror, and it can be expressed as
(23)$$PST(\varphi )=S(\varphi )\sum_{i=1}^{n}BSDF{\left(\varphi \right)}_{i}\Omega {\left(\varphi \right)}_{i}$$
where *n* is the number of mirrors the incident light reaches, which is not equal to the number of mirrors in the optical system. Combined with the determination condition, a loop-nested program was designed to plot the PST curve of the optical system, as shown in Figure 9 below. 

According to the PST model and iteration loop, the multiple reflections are simulated by calculating the reflective angle and ray height in the optical system, which were deduced from the previous formulations. If the light meets the requirement (like the stray light passing through a line segment), it will jump to another algorithm and change the calculation step to simulate the reflection or change. When the light meets the requirement to stop, it will jump out of the loop and over the manipulation, which means the light will arrive at the detector or run out of the calculation range.

Since the sine function value used in BSDF has π/2 symmetry, it is possible to generate protrusions even when the scattering and reflection angles are not identical. The model in this paper mainly reflects the ability of the structural characteristics of the optical system to suppress stray light, after which this paper processed the protrusion data created by the formula as well as the data of the corresponding values of the angle of incidence independent of the specular surface. The comparison between the program prediction results and the TracePro results was conducted to verify its validity.

## 4. Comparative Verification of the Model Fitting Validity and Screening of the Initial Structure

### 4.1. Verification of the Determination Condition

The verification was conducted by the optical payload of the Spectrum 02 Satellite. This optical system is an off-axis three-mirror reflective optical system with a field of view angle of 6.4° × 2.63°.

The surface-scattering property settings for the optical payload were derived from the measured BSDF data, which were fitted using the ABg model. The light source plane was set directly above the detector, tracing 1.2 million rays, and a light barrier was set outside the normal imaging path with perfect absorption properties, as shown in Figure 10 below. The red light is normal light and the blue light is the scatter light. At an off-axis angle of 39°, the rays outside the field of view reached the tertiary mirror and produced a once-scattered attenuation stray light that affected the detector.

When using the rapid evaluation method, only a small number of parameters need to be entered into the program to obtain results, including the parameters of the optical system (radii, distance, field of view, ABg model’s coefficient of each mirror, and light source’s size); these parameters can also be used to calculate the angle of radiation acceptance coefficient and the determination condition, according to the formulation in Section 3. Based on the analysis of the determination condition in this study, it can be seen that the system has scattered stray light at an incidence angle of 39°.

The magnitude of the stray light received by the detector was 2.6768 × 10^−5^ W/m^2^, as shown in Figure 11a. The simulation results obtained by the fast evaluation model have a protrusion of 2.5 × 10^−5^ W/m^2^ in the 39° angular range, and the predicted results are close to the tracing results with excellent fitting results, verifying that the model can quantitatively predict the direct-incident stray light and once-scattered attenuation stray light outside of the field of view.

### 4.2. System Fitting Results

Based on the parameter settings in the previous section, we continued to use TracePro to plot the PST curve at a full angle, and the results are shown in the black curve in the figure below. According to the fast stray light performance evaluation prediction model, the results of the PST curve are as Figure 12.

The PST curve shows that the PST value rises slightly above the E-5 order of magnitude at an off-axis angle of 40° in the Y field-of-view direction, corresponding to the effect of the strong stray light analyzed above. And there is a stopping point in the decline near 20°.

A comparison of the fitting results shows that there is a good fit over the range of the radiant reception angles, with comparable magnitudes of the predicted stray light, with the curves roughly in the right direction, and with a successful prediction of the out-of-field-of-view stray light. In the case of the system used in this paper, it took only 0.000055 s to produce the results, avoiding the time-consuming nature and complexity of the traditional TracePro working in tandem with the rest of the programming software.

### 4.3. Screening of the Initial Structures with a Stray Light Suppression Ability

This fast evaluation method based on the structure property and specular scattering property of the optical system can be used to find the initial structure of the optical system with a stray light suppression capability. For the above optical system, under the same design index and specular scattering property, the structure of the optical system with different radii and spacing was designed, as shown in Figure 13. The main difference between the initial structures is the distance between the primary mirror and secondary mirror. Detailed parameters are shown in Table 1.

From structure 1 to structure 4, the incident light source surface was set at the farthest mirror or detector at the same side. The PST values of the various structures were obtained by the fast stray light evaluation method as Figure 14:

As it can be observed in the figure, the initial structure 4 has no peaks of out-of-field stray light protrusions, which indicates that this system structure has the effect of suppressing the once-scattered attenuation stray light caused by the out-of-field stray light. Moreover, the volume of the initial structure 4 is smaller than that of the others, and this method has the potential to find the optical structure with the optimal stray light suppression ability.

The above four initial structures were subjected to basic photomechanical modelling and Monte Carlo ray tracing was performed using TracePro at large angular intervals to verify the validity of the screening and to save time at the same time. After 4 h of ray tracing, the following Figure 15 was obtained: Structures 1 to 3 have protrusions in the descent process, i.e., the once-scattered attenuation stray light arriving at the image plane caused by the stray light from outside of the field of view reaching the tertiary mirror. The results are consistent with those of the rapid evaluation method, proving the effectiveness of this method for structure screening.

## 5. Conclusions

In this study, a fast quantitative evaluation model and a determination condition for stray light passage were proposed to address the complexity of and large time consumed by the traditional evaluation method of stray light suppression ability. This paper introduced the decenter and tilt, supplemented and extended the application of light vector transfer in a non-coaxial near-linear system, introduced the radiation-receiving coefficients based on the radiative transfer theory of BSDF characteristics, and combined the structural parameters of the optical system. It investigated the transmission characteristics of stray radiation in optical systems with different structural forms using vector light transfer and established a model for evaluating the stray light suppression ability of optical systems from the perspective of the global theory. It effectively replaced the millions of ray tracing within the range of the specular radiation reception angle. The three-dimensional system was simplified to a two-dimensional model, which is feasible in the system with a better symmetry. A foundation was laid for the subsequent search for an optimal stray light elimination structure. This study is of great significance for the development of a lightweight and compact space camera with a low stray light energy to improve the radiometric accuracy of imaging.

## Figures and Tables

**Figure 1 sensors-23-09182-f001:**
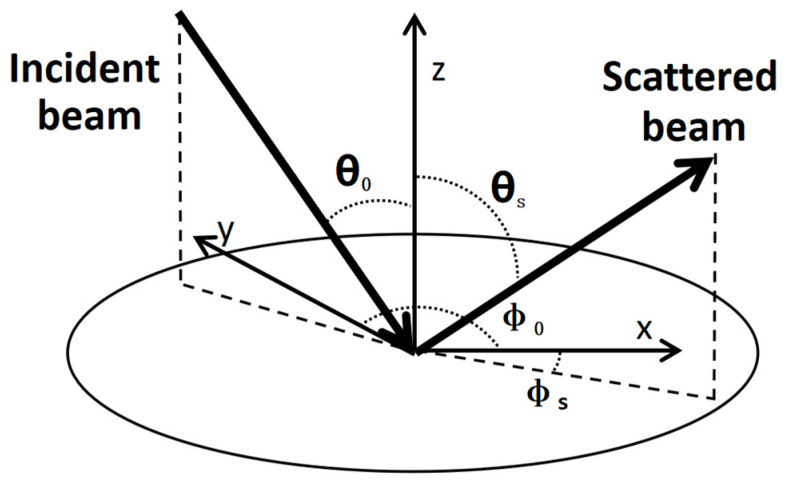
Angles used in the definition of the BRDF.

**Figure 2 sensors-23-09182-f002:**
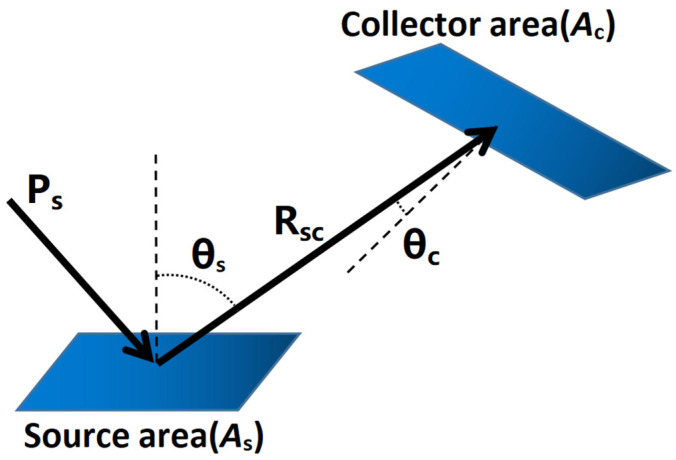
Schematic diagram of basic radiative transfer.

**Figure 3 sensors-23-09182-f003:**
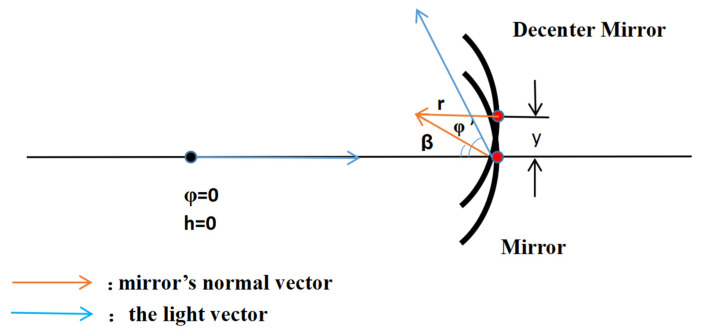
Graphical representation of the decenter of the mirror surface.

**Figure 4 sensors-23-09182-f004:**
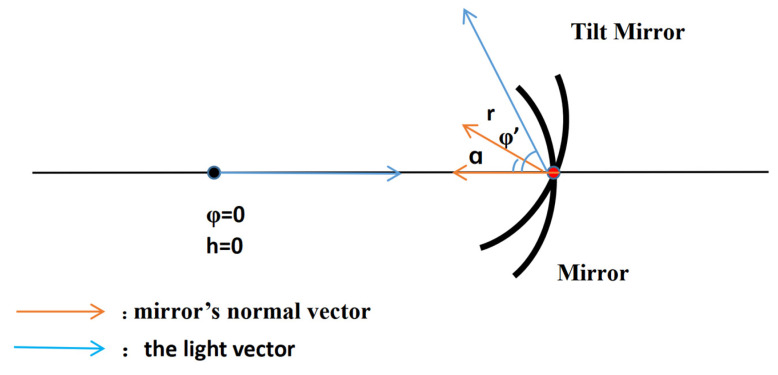
Graphical representation of the tilt of the mirror surface.

**Figure 5 sensors-23-09182-f005:**
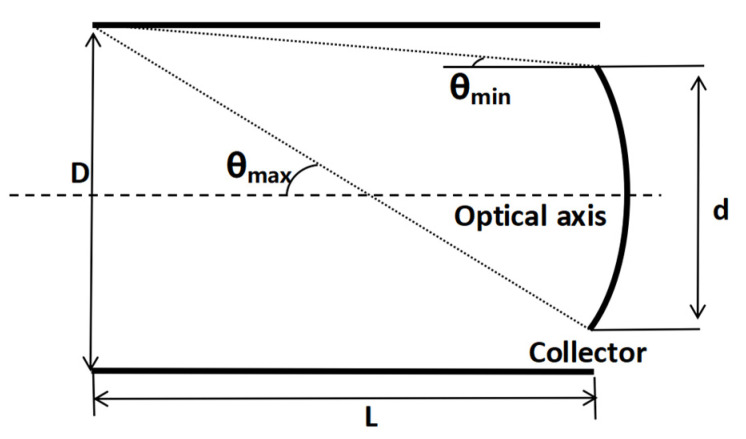
Schematic diagram of the radiation-receiving coefficient.

**Figure 6 sensors-23-09182-f006:**
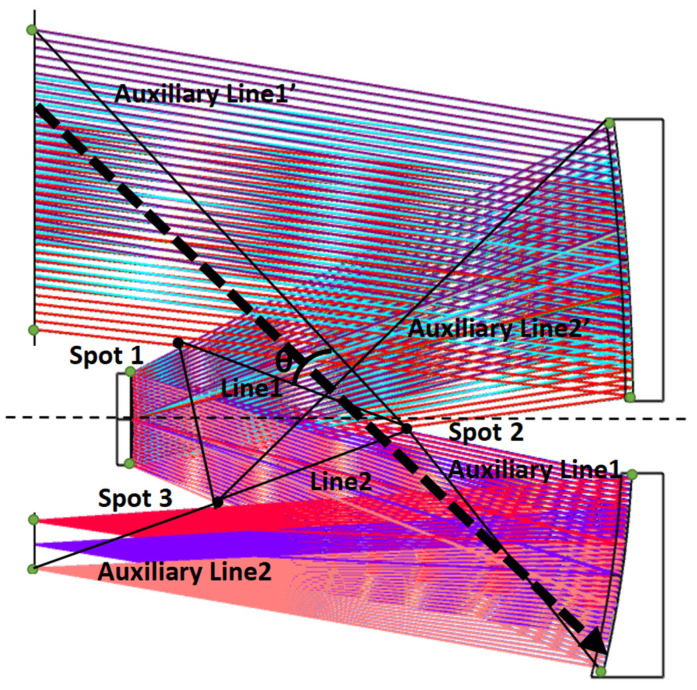
Schematic diagram of the stray light passing through the line segment.

**Figure 7 sensors-23-09182-f007:**
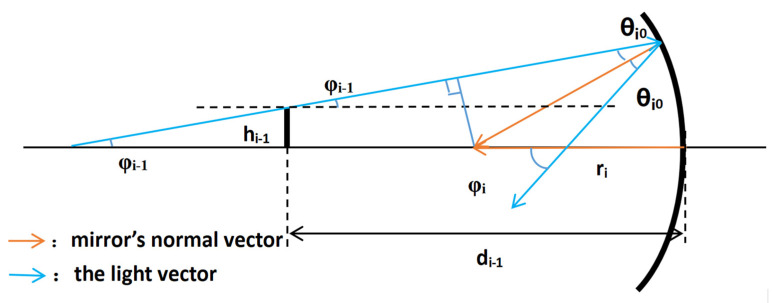
Graphical representation of the incident light versus the angle of reflection.

**Figure 8 sensors-23-09182-f008:**
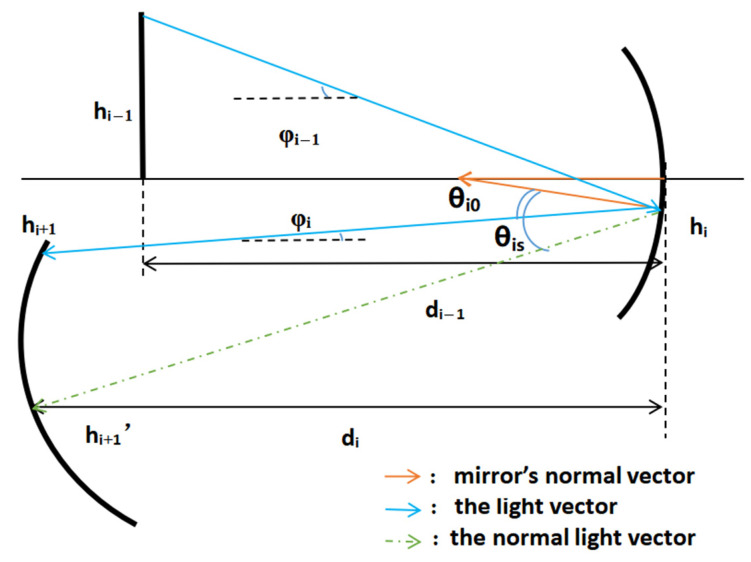
Schematic diagram of the radiative transfer of non-normal incident light.

**Figure 9 sensors-23-09182-f009:**
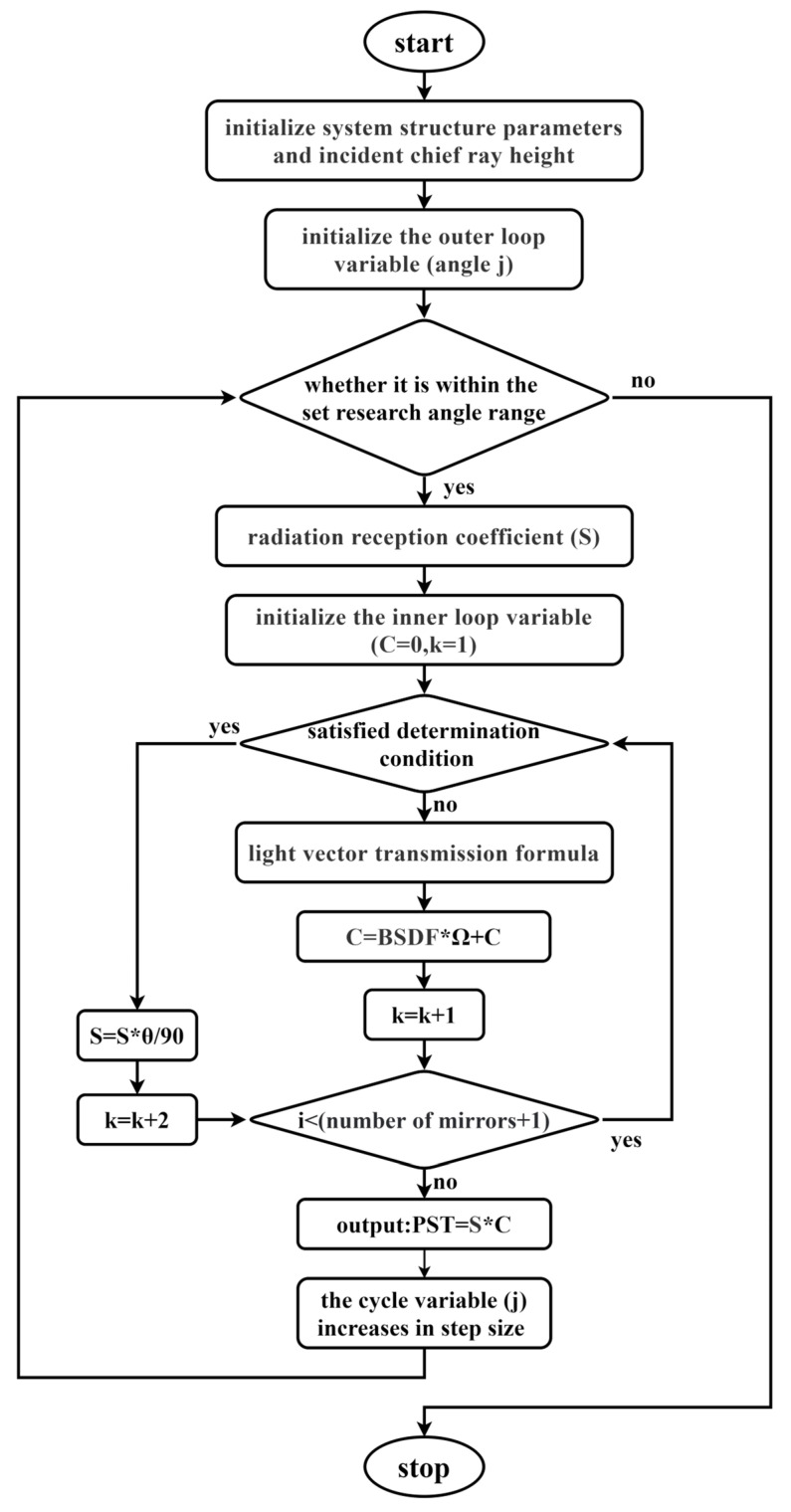
Flow chart of the main loop-nested program.

**Figure 10 sensors-23-09182-f010:**
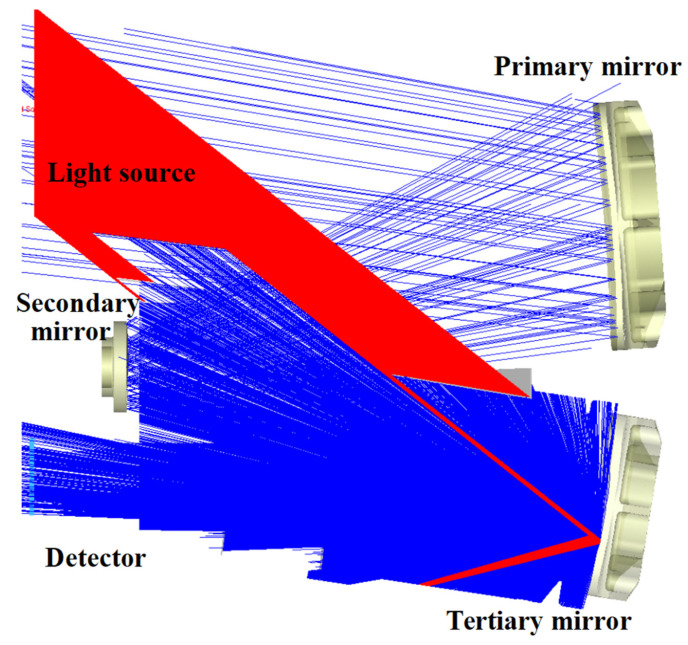
Simulation of ray tracing at a 39° angle of incidence.

**Figure 11 sensors-23-09182-f011:**
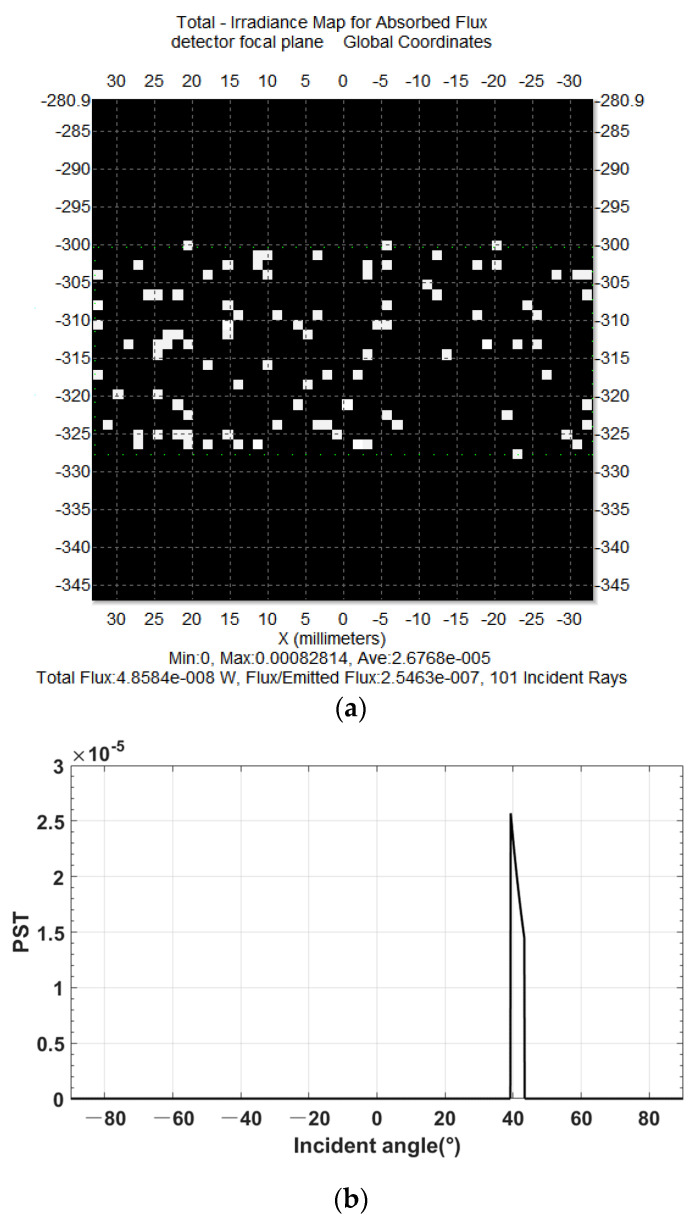
Comparison of the prediction results of the determination conditions with the simulation results. (**a**) Detector image plane average irradiance by TracePro. (**b**) The out-of-field-of-view stray light results of the fast stray light performance evaluation.

**Figure 12 sensors-23-09182-f012:**
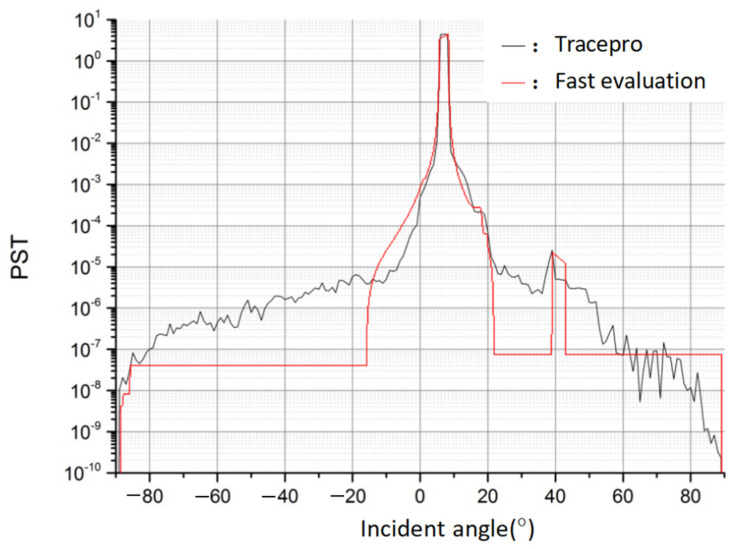
Comparison of the TracePro results with the fast stray light performance evaluation results.

**Figure 13 sensors-23-09182-f013:**
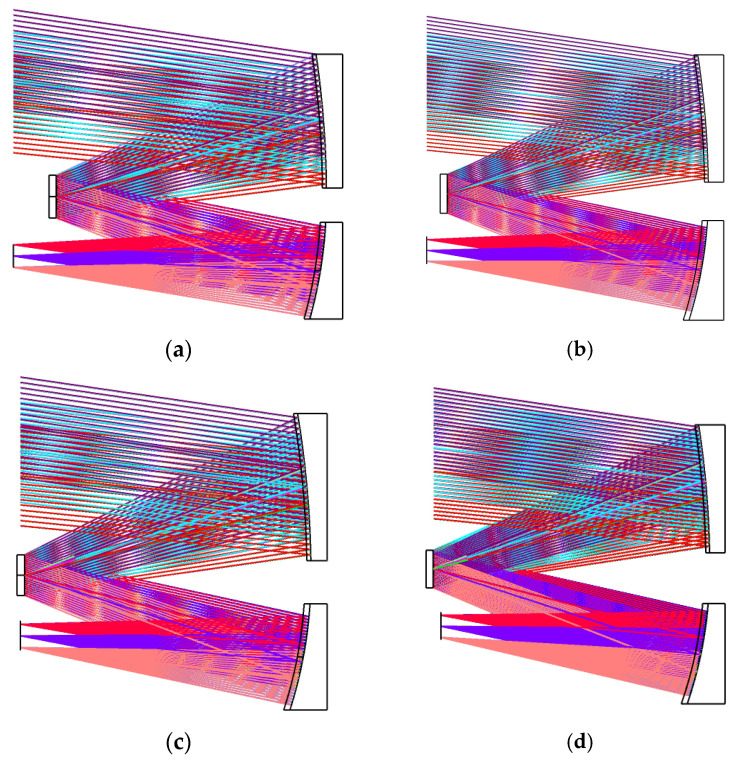
Different initial structures. (**a**) Initial structure 1; (**b**) initial structure 2; (**c**) initial structure 3; and (**d**) initial structure 4.

**Figure 14 sensors-23-09182-f014:**
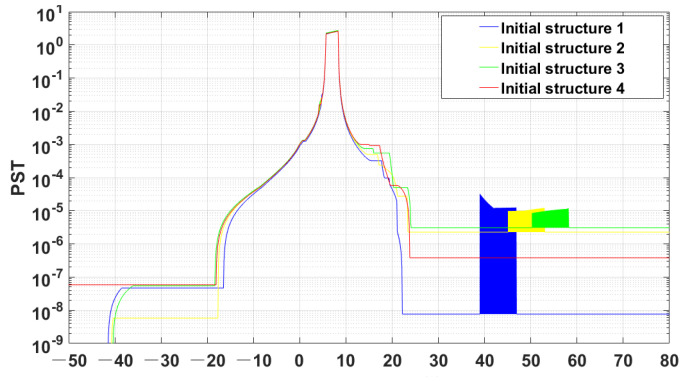
Comparison of the PST for each structure based on the rapid evaluation method.

**Figure 15 sensors-23-09182-f015:**
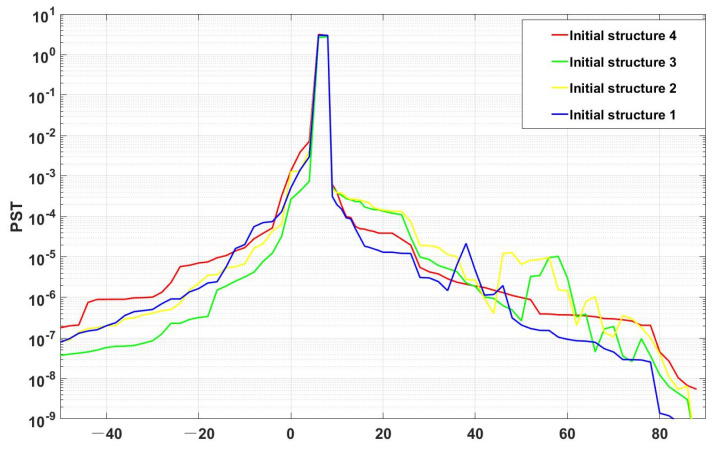
Analysis results of the initial structures using TracePro.

**Table 1 sensors-23-09182-t001:** Initial structure.

	Radii of Primary Mirror	Primary and Secondary Mirror Spacing	Radii of Secondary Mirror	Secondary and Tertiary Mirror Spacing	Radii of Primary Mirror	Tertiary Mirror and Image Plane Spacing	Volume
Initial structure1	−1204.92	−335.00	−391.10	335.00	−569.55	−389.22	3.11 × 10^7^
Initial structure2	−1184.73	−340.00	−381.20	340.00	−552.95	−367.34	3.15 × 10^7^
Initial structure3	−1193.64	−350.00	−378.06	350.00	−544.18	−355.83	3.22 × 10^7^
Initial structure4	−1211.67	−360.00	−377.98	360.00	−540.43	−350.79	3.09 × 10^7^

## Data Availability

Data are contained within the article.

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
