# Peer review of "Fast Stray Light Performance Evaluation Based on BSDF and Radiative Transfer Theory"

_sensors, 2023, doi:10.3390/s23229182_

Round 1

Reviewer 1 Report

Comments and Suggestions for Authors

This manuscript presents the analytic method to evaluate the stray light performance. Since this method is very fast compared to the Monte Carlo method, the optimization of baffle design would be very efficient. I would like to recommend to publish this work, after concerning the following issues:

(1) The introduction section need to review the straylight analysis methods in the literature. And some basic definitions in equations 1, 2, and 3 can be moved to Section 2.

(2) The typical stray light simulations considers multiple reflections as the light pass into the optical system. The authors need to explain how they manipulated this multiple reflections in their calculations. This can be done with the iteration loop in Figure 9.

(3) In Section 4, the detailed input parameters for the TracePro method and the radiative transfer method. Because the optical structure model of the radiative transfer method are simplified, the way of using the input parameters should be discussed.

(4) The Figure 15 shows the suppressed out-of-field stray light in the Initial structure4. The same PST from the TracePro should be included to confirm the method.

(5) The reference numbers should be checked.

(6) The numerical values in the Table have too many significant figures. The values should have only meaningful significant figures.

(7) Figure 11 should have units (e.g., the x-axis in Figure 11-b) and clear figure captions to explain what to compare these methods.

(8) Figure 13 can be plotted over Figure 12 to easily compare the results. The figure captions should be revised. 

(9) The optical structures in Figure 14 looks very similar each other. There should be notes to show the differences.

=== End of comments ===

Comments on the Quality of English Language

The names in the figure captions need to be clear. For example, Figure 12 shows "Trace Pro ray tracing PST curve" and Figure 13 shows "PST model fitting curves" which use different names for same concepts.

Reviewer 2 Report

Comments and Suggestions for Authors

The authors demonstrate a new mathematical model for stray light characterization in reflective off-axis illuminated cameras, widely used in aerospace. Authors verify model validity by simulating stray light suppression of 3-mirror reflective optical system with known stray light suppression parameters. The model is benchmarked against proven computation-heavy model based on Monte Carlo and ray tracing method, implemented in TracePro software package. Developed model allows rapid stray light analysis with high precision, assuming approximation where optical system is symmetrical and reduced to 2D geometry.

I recommend the article for publication after my minor comment is addressed:

1) Line 247: authors briefly mention mirror surface roughness affects stray light profile of optical systems. Would be nice to add couple simulation examples or a discussion, comparing stray light from mirrors with different surface roughness values. This is of interest since cost of higher quality mirror surface grows exponentially, and finding a good cost/stray light compromise is important.

Round 2

Reviewer 1 Report

Comments and Suggestions for Authors

I appreciate all your efforts to revise the manuscript.

Comments on the Quality of English Language

N/A